# Activated Protein C Ameliorates Tubular Mitochondrial Reactive Oxygen Species and Inflammation in Diabetic Kidney Disease

**DOI:** 10.3390/nu14153138

**Published:** 2022-07-29

**Authors:** Rajiv Rana, Jayakumar Manoharan, Anubhuti Gupta, Dheerendra Gupta, Ahmed Elwakiel, Hamzah Khawaja, Sameen Fatima, Silke Zimmermann, Kunal Singh, Saira Ambreen, Ihsan Gadi, Ronald Biemann, Shihai Jiang, Khurrum Shahzad, Shrey Kohli, Berend Isermann

**Affiliations:** Institute of Laboratory Medicine, Clinical Chemistry and Molecular Diagnostic, University Hospital Leipzig, 04103 Leipzig, Germany; rajiv.rana@medizin.uni-leipzig.de (R.R.); jayakumar.manoharan@medizin.uni-leizig.de (J.M.); anubhuti.gupta@medizin.uni-leipzig.de (A.G.); dheerendra.gupta@medizin.uni-leipzig.de (D.G.); ahmed.elwakiel@medizin.uni-leipzig.de (A.E.); hamzah.khawaja@medizin.uni-leipzig.de (H.K.); sameen.fatima@medizin.uni-leipzig.de (S.F.); silke.zimmermann@medizin.uni-leipzig.de (S.Z.); kunal.singh@medizin.uni-leipzig.de (K.S.); saira.ambreen@medizin.uni-leipzig.de (S.A.); ihsan.gadi@medizin.uni-leipzig.de (I.G.); ronald.biemann@medizin.uni-leipzig.de (R.B.); shihai.jiang@medizin.uni-lepzig.de (S.J.); khurrum.shahzad@medizin.uni-leipzig.de (K.S.); shrey.kohli@medizin.uni-leipzig.de (S.K.)

**Keywords:** diabetic nephropathy, activated protein C, renal inflammation, inflammasome, reactive oxygen species

## Abstract

Diabetic kidney disease (DKD) is an emerging pandemic, paralleling the worldwide increase in obesity and diabetes mellitus. DKD is now the most frequent cause of end-stage renal disease and is associated with an excessive risk of cardiovascular morbidity and mortality. DKD is a consequence of systemic endothelial dysfunction. The endothelial-dependent cytoprotective coagulation protease activated protein C (aPC) ameliorates glomerular damage in DKD, in part by reducing mitochondrial ROS generation in glomerular cells. Whether aPC reduces mitochondrial ROS generation in the tubular compartment remains unknown. Here, we conducted expression profiling of kidneys in diabetic mice (wild-type and mice with increased plasma levels of aPC, APC^high^ mice). The top induced pathways were related to metabolism and in particular to oxidoreductase activity. In tubular cells, aPC maintained the expression of genes related to the electron transport chain, PGC1-α expression, and mitochondrial mass. These effects were associated with reduced mitochondrial ROS generation. Likewise, NLRP3 inflammasome activation and sterile inflammation, which are known to be linked to excess ROS generation in DKD, were reduced in diabetic APC^high^ mice. Thus, aPC reduces mitochondrial ROS generation in tubular cells and dampens the associated renal sterile inflammation. These studies support approaches harnessing the cytoprotective effects of aPC in DKD.

## 1. Introduction

Due to the pandemic-related incidence of obesity, resulting largely from unbalanced nutrient intake and energy expenditure, diabetes mellitus and its associated complications are increasing worldwide [1,2]. Diabetic kidney disease (DKD) is a major microvascular complication of diabetes mellitus. DKD is currently the most frequent cause of end-stage renal disease. In addition, DKD increases the risk of cardiovascular disease and associated morbidity and mortality. In recent years, new therapeutic options for DKD have evolved, such as SGLT-2 inhibitors, DPP-4 inhibitors and GLP-1 receptor agonists, and the incidence of DKD and associated vascular complications continue to increase [3,4,5]. Hence, a better understanding of DKD is required to define new therapeutic approaches.

DKD is a diabetic microvascular complication associated with sterile inflammation. Endothelial dysfunction with loss of endothelial nitric oxide synthase (eNOS) or endothelial thrombomodulin contributes to diabetes-associated vascular diseases [6,7]. Thrombomodulin is a transmembrane protein that is mostly expressed in endothelial cells. Upon the binding of thrombin, the complex of thrombomodulin and thrombin activates zymogen protein C, resulting in the generation of anticoagulant and cytoprotective protease-activated protein C (aPC) [8]. We and others have demonstrated the cytoprotective effects of aPC in various disease models [6,9,10,11]. Importantly, the cytoprotective effects of aPC are at least partially independent of its anticoagulant function [12]. Hence, aPC mimetics that imitate aPC signaling properties without interfering with the coagulation system have been developed, and some have been clinically tested [13]. These pharmaceutical substances are promising therapeutics.

With regard to DKD, we and others have demonstrated that aPC protects against experimental DKD [6,14,15]. Considering that DKD is mostly viewed as a glomerular disease, we and others focused on the glomerular compartment in these studies. In particular, we were able to demonstrate that aPC reduces excess mitochondrial reactive oxygen species (ROS) in glomerular cells [10]. However, it is increasingly recognized that DKD reflects both glomerular and tubular damage [16,17]. As tubular cells are rich in mitochondria, we hypothesized that aPC likewise targets mitochondrial function in the tubular compartment. Hence, in the current study, we aimed to define whether aPC protects not only the glomerular but also the tubular compartment from excessive mitochondrial ROS generation.

## 2. Materials and Methods

### 2.1. Reagents

The following antibodies were used in the current study: mouse anti-β-actin (Cell Signaling Technology), NF-κB-p65 (Cell Signaling Technology), phospho-NF-κB-p65 (Cell Signaling Technology, Frankfurt, Germany), Caspase-1 (Santa Cruz, Heidelberg, Germany), Asc (Abcam, Cambridge, UK), Nlrp3 (Novus Biologicals, Centennial, USA), F4/80 (R&D Systems, Wiesbaden, Germany), nitrotyrosine (Cell Signaling Technology), 8-oxo-dG (R&D Systems), goat anti-rabbit IgG HRP (Cell Signaling Technology), and anti-mouse IgG HRP (Cell Signaling Technology). The following secondary antibodies were used for immunofluorescence: Texas Red Rabbit Anti-Mouse IgG (Vector Laboratories, Newark, CA, USA), Alexa Fluor 466 (Vector Laboratories, USA), and Alexa Fluor 568 (Vector Laboratories, USA). The following antibodies were used for FACS: MitoTracker Green (Invitrogen, Karsruhe, Germany) and MitoSox (Invitrogen). Other reagents were as follows: BCA reagent (Perbio Science, Bonn, Germany); ProLong™ Gold mounting medium with DAPI (Invitrogen); nitrocellulose membrane (Bio-Rad, Feldkirchen, Germany) and Immobilon transfer membrane western chemiluminescent HRP substrate (Merck, Millipore, Darmstadt, Germany); haematoxylin Gill II, acrylamide, agarose (Carl ROTH, Karlsruhe, Germany); aqueous mounting agent (ZYTOMED, Bargteheide, Germany); PBS (Life Technologies, Schwerte, Germany); Rompun 2% (Bayer, Leverkusen, Germany); and ketamine 10% (beta-pharm, Germany).

### 2.2. Mice

Wild-type C57BL/6 mice were obtained from Janvier, France. Mice were backcrossed to the C57BL/6J background for at least 10 generations. APC^high^ mice have been described previously [10]. German guidelines for animal experiments were followed after approval by the local Animal Care and Use Committee (Landesverwaltungsamt Halle, Germany. Sustained hyperglycaemia in eight-week-old mice was induced by the intraperitoneal administration of streptozotocin (STZ) (60 mg/kg) dissolved in 0.05 M sterile sodium citrate, pH 4.5, for five consecutive days. The mice were classified as diabetic when the blood glucose level was above 300 mg/dL. Blood glucose levels were measured with ACCU-CHEK glucose sticks from tail vein blood. Blood glucose levels were maintained with insulin to avoid fatal hyperglycaemia, as described [18].

### 2.3. Determination of Albuminuria

Albumin and creatinine levels in the urine of mice were measured as previously described [19,20]. Mice were placed in individual metabolic cages for 12 h to collect urine samples after 24 weeks of STZ-induced hyperglycaemia. Albuminuria was determined using a mouse albumin enzyme-linked immunosorbent assay kit (Bethyl Laboratories) according to the manufacturer’s instructions. Urinary creatinine was determined using the Cobasc501 module assay from Roche Diagnostics [14].

### 2.4. Cell Culture

Boston University mouse proximal tubular cells (BUMPT) were cultured in DMEM containing low glucose (5 mM), 10% FBS, 5000 units of interferon-y, and 1% penicillin and streptomycin at 33 °C as previously described [21]. Proliferating BUMPT cells were trypsinized (0.05%) and divided. To induce differentiation, BUMPT cells were incubated at 37 °C without interferon-Y and with 5% CO_2_ in a humidified chamber.

### 2.5. RNA Expression Profiling

Expression profiling from the type-I hyperglycaemic mouse model was performed using the Affymetrix mouse Gene 1.0 ST GeneCHIP array. The raw expression data were normalized at the gene level with the RMA (Robust Multiarray Analysis) algorithm (oligo package in R), batch corrected with ComBat (as the measurement was done in batches), annotated with MoGene-1_0-st-v1.na32.mm9.transcript.csv and filtered for control probe sets (arriving at 28,440 annotated genes/transcripts). The statistical analysis was performed on the filtered and annotated data with the Limma package (moderated t statistics). To correct for multiple testing errors, the Benjamini–Hochberg correction (BH) was used. The threshold for identifying differentially expressed genes (DEGs) was set at a logFC value of ±0.58, corresponding to a 1.5-fold change in expression. Statistically significant DEGs (*p* < 0.01; FDR < 0.1) were used for functional annotation. Identified DEGs were functionally annotated using KEGG analyses via the Gene Set Enrichment Analysis (GSEA) database.

### 2.6. Flow Cytometry

To assess mitochondrial content, mouse proximal tubular cells (BUMPT) were stained with the mitochondrial probe MitoTracker Green (Life Technologies). BUMPT cells (5 × 105 cells) were incubated with 100 nM MitoTracker Green for 30 min at 37 °C. In another set of experiments, BUMPT cells were incubated with 1 µM MitoSOX™ Red for 30 min at 37 °C. After washing, the cells were resuspended in FACS buffer and subjected to FACS on an Attune NxT (Thermo Scientific). MFI (median fluorescence intensity) was measured using FlowJoTM v10.8.1 software. First, live cell, population gated (SSC-A vs. FSC-A), followed by singlets selection using FSC-H vs. FSC-A. Finally, the statistical MFI function for the respective channels, i.e., PE for MitoSox and FITC for MitoGreen, was applied independently to all groups. This function provides the MFI for individual samples.

### 2.7. Immunoblotting

Immunoblotting was performed as previously described [18]. Cell protein lysates were made using an ice-cold RIPA lysis buffer (50 mM Tris-pH 7.4, 1% NP-40, 0.25% sodium deoxycholate, 150 mM NaCl, 1 mM EDTA, 1 mM Na3VO4, 1 mM NaF) containing a protease and phosphate inhibitor cocktail. The protein lysate was centrifuged at 12,000× *g* for 10 min at 4 °C, the supernatant was collected, and the protein content was quantified using a BCA reagent (Thermo Fisher Scientific, Dreieich, Germany). Briefly, 20 µg of protein was loaded onto an SDS polyacrylamide gel, electrophoretically separated, and transferred to a PVDF membrane. The PVDF membranes were incubated overnight with primary antibodies in a blocking solution (1× TBS with 0.1% Tween-20 and 5% BSA). The next day, the PVDF membrane was washed three times for 10 min at RT in TBST (1× TBS with 0.1% Tween-20), and the corresponding secondary horseradish peroxidase-conjugated antibodies (1:2000) were added for 2 h at RT. Blots were analysed using the Azure System (Azure Biosystems) chemiluminescence reagent. ImageJ was used to measure the protein band density. β-Actin was used as a reference gene for immunoblotting.

### 2.8. Quantitative Polymerase Chain Reaction (qPCR)

RNA was isolated from BUMPT cells using the TRIzol method (Life Technologies, Darmstadt, Germany) according to the manufacturer’s instructions. RNA quality and integrity were ensured by the A260/280 ratio and agarose gel electrophoresis, respectively. One microgram of RNA was used for the reverse transcription reaction. DNase 1 (1 U/1 µg RNA) treatment was performed for 30 min at 37 °C. Reverse transcription was performed using the RevertAid First Strand cDNA Synthesis Kit (Thermo Fisher Scientific). qRT–PCR was performed using an Applied Biosystems ViiA 7 Real-time System (Thermo Fisher Scientific, Dreieich, Germany), SYBR Green (Takyon™, Eurogentec, Cologne, Germany), and primers (Appendix A). For quantitative analysis, the results were normalized to 18S RNA. The ΔΔCt method was used to analyze relative gene expression [22].

### 2.9. Total DNA Isolation

Total DNA collection and relative expression levels of mtDNA/nDNA using qPCR were performed according to a previous study [23]. Briefly, cells were lysed and treated with RNase A (100 µg/mL) to eliminate the presented RNA. DNA were precipitated by ammonium acetate (7.5 M) and isopropanol (0.7 *v*/*v*). A comparison of ND2 expression (from the mitochondrial genome) relative to HK1 expression (from the nuclear genome) were selected to evaluate the relative copy number of mtDNA to nDNA (Appendix A). QPCR was performed using FastStart Universal SYBR Green Masters (Roche) ac-cording to the manufacturer’s instructions. The ΔΔCt method was used to calculate the mtDNA/nDNA ratio.

### 2.10. ATP Level Quantification

Intracellular ATP levels of BUMPT cells were measured using a CellTiter-Glo^®^ 2.0 assay (Promega, Walldorf, Germany). Briefly, cells were treated with high glucose (25 mM) with or without aPC pretreatment (20 nM, 1 h, HG + aPC) for 24 h and then the cell number was measured by Hoechst 33,342 dye at 350/462 nm for normalization. Further cells were treated with CellTiter-Glo reagent and incubated at room temperature for 10 min, and luminescence was measured using a Cytation 5 Imaging reader.

### 2.11. Histochemistry

Kidney tissue from the mice was processed as described [24]. In brief, mice were perfused with ice-cold PBS. Kidney tissue was then fixed in 4% buffered formalin for 24 h before being embedded in paraffin and cut into 5 µm sections [24]. The kidney sections were deparaffinized in xylene and rehydrated in a graded ethanol solution before periodic acid-Schiff staining (PAS) according to the current DCC (Diabetes Complications Consortium) protocol (https://www.diacomp.org/ (accessed on 1 June 2022)).

### 2.12. Immunofluorescence

For immunofluorescence, the hydrated kidney paraffin sections were blocked with 5% donkey serum in PBS-Tween (PBST) for 1 h in a humidified chamber after antigen retrieval. After incubation overnight at 4 °C with primary antibodies, the sections were washed three times for 10 min in 1× PBS and then incubated with the appropriate secondary fluorescently labelled antibodies (1:400). After washing three times in 1X PBS for 10 min each, DAPI nuclear counterstaining was performed. Images were acquired using a Keyence fluorescence microscope with the same settings. To measure the fluorescence intensity of immunofluorescent images, the image of the corresponding channel was opened in Image J software. The integrated intensity was selected and the intensity of the entire image for the respective channel was measured. Ten visual fields per slide were randomly selected from each section and measured. The average intensity was calculated for each animal.

### 2.13. Transmission Electron Microscopy

Transmission electron microscopy was used for ultrastructural analyses of the glomerular basement membrane (GBM) as previously described [25].

## 3. Results

### 3.1. aPC Regulates Pathways Related to Metabolism and Oxidoreductase Activity in DKD

To study pathways regulated by aPC in diabetic nephropathy, we compared nondiabetic (C) and diabetic (DM) wild-type (WT) mice to mice with increased plasma levels of activated protein C (APC^high^). Consistent with previous data [25], albuminuria and glomerular damage in diabetic APC^high^ mice were markedly reduced compared to those in WT-DM mice and were not different from those in WT control mice (Appendix A). To gain insight into the possible pathways regulated by aPC in DKD, we performed bulk RNA sequencing and compared total RNA from APC^high^-DM mice and WT-DM mice after 24 weeks of hyperglycaemia. A total of 1747 genes (965 induced, 782 repressed) were differentially expressed (Figure 1a) (Appendix A). Analyses using the annotation, visualization, and integrated discovery database (DAVID) combined with KEGG gene annotation for induced genes revealed that metabolic pathways were significantly enriched in APC^high^-DM mice compared with WT-DM mice (Figure 1b). Selected genes with FDR ≥ 0.05 and related to metabolic and inflammatory pathways are labelled in the volcano plot (Figure 1a). Among the genes associated with metabolic pathways, most genes were related to oxidoreductase activity (Figure 1c). These data suggest that aPC regulates metabolic pathways and particularly the redox activity in DKD.

### 3.2. aPC Protects against Mitochondrial Dysfunction in DKD

As we conducted bulk sequencing, changes in gene regulation likely reflect changes in tubular cells, which constitute approximately 80% of renal cells. Accordingly, to determine the effect of aPC on mitochondria, we used tubular cells for in vitro experiments. Exposing murine tubular cells to high glucose (25 mM, 24 h) induced the expression of various genes involved in the electron-transport chain (ETC), implying mitochondrial dysfunction. Pretreatment of murine tubular cells with aPC (20 nM, 1 h) prevented the high glucose-induced induction of genes related to the ETC (Figure 2a). To analyse mitochondrial biomass, we performed FACS using MitoTracker Green, which accumulates in mitochondrial lipid membranes. Exposing murine tubular cells (BUMPT) to high glucose (25 mM, 24 h) reduced the accumulation of MitoTracker Green, suggesting a loss of mitochondrial biomass. This glucose-induced effect was prevented by pretreatment with aPC (20 nM, 1 h, Figure 2b), corroborating the protective effect of aPC on mitochondria. We next studied the effect of aPC on PGC-1α, a master regulator of mitochondrial biosynthesis, which is abundantly expressed in renal tubular epithelial cells [26]. Tubular PGC-1α expression, as determined by immunohistochemistry, was reduced in 24-week-old diabetic mice compared to nondiabetic control mice. In diabetic APC^high^ mice, PGC-1α expression was partially maintained compared to nondiabetic WT mice (Figure 2c,d). To further analyse mitochondrial dynamics, we analyzed mRNA expression of DRP1 and OPA1 in control, high glucose, and high glucose with aPC exposed cells. DRP1, as a marker of mitochondrial fission, and OPA1, as a marker of mitochondrial fusion, convey important functions in mitochondrial biogenesis. The high-glucose induced changes were reversed upon exposure of cells to aPC (Figure 2e,f). The protection of mitochondrial biomass and PGC-1α by aPC in DKD suggests that aPC maintains mitochondrial function in tubular cells despite persistently elevated glucose levels.

### 3.3. aPC Reduces Tubular ROS in Diabetic Nephropathy

Given the above results showing the regulation of pathways related to oxidoreductase activity and the protection of mitochondrial markers by aPC despite high glucose, we next investigated the effect of aPC on high glucose-induced ROS production in tubular cells. First, the exposure of mouse tubular cells (BUMPT) to a high glucose concentration (25 mM, 24 h) induced ROS formation, as reflected by increased MitoSox staining. Pretreatment with aPC (20 nM, 1 h) before exposing cells to high glucose prevented increased ROS formation (Figure 3a), indicating that aPC reduces high glucose-induced ROS production in tubular cells. In addition, aPC prevented high glucose-induced ATP levels and restored mitochondrial DNA content in tubular cells (Figure 3b,c). To determine whether aPC has an effect on tubular ROS production in vivo, we studied reactive metabolites generated upon increased ROS generation. First, we stained renal sections for 8-oxo-dG, a marker for ROS-induced DNA damage, together with the mitochondrial marker VDAC. While in nondiabetic control mice, 8-oxo-dG staining was low and in particular did not colocalize with the mitochondrial marker VDAC, staining for 8-oxo-dG was induced and prominently colocalized with VDAC in tubular cells of diabetic mice (Figure 3d,e). In contrast, in diabetic APC^high^ mice, both the staining intensity for 8-oxo-dG and colocalization with VDAC were markedly reduced. These results indicate that ROS induce DNA damage in tubular cells of diabetic mice, particularly in mitochondria, which, however, can be prevented by aPC. Next, we determined the reactive metabolite peroxynitrite (ONOO-). Compared to nondiabetic wild-type control mice, staining for nitrotyrosine was markedly enhanced in the tubular compartment of diabetic wild-type mice. Increased nitrotyrosine staining was not apparent in diabetic APC^high^ mice (Figure 3f,g). Increased ROS generation drives NF-ĸB signalling, which is closely linked to diabetic complications, including DKD [27,28,29]. To determine whether aPC dampens ROS-induced NF-ĸB activation, we stimulated mouse tubular cells (BUMPT) with H_2_O_2_ (100 µM, 24 h), which induced phosphorylation of the p65 NF-ĸB subunit, reflecting NF-ĸB activation (Figure 3h,i). Pretreatment of BUMPT cells with the mitochondrial ROS scavenger MitoTempo (10 µM, 1 h) or with aPC (20 nM, 1 h) prevented ROS-induced p65 NF-ĸB phosphorylation (Figure 3h,i). Taken together, these results demonstrate that aPC reduces high glucose-induced tubular ROS generation.

### 3.4. aPC Reduces Inflammasome Activation in Diabetic Nephropathy

Increased ROS and NF-ĸB activity are known to induce Nlrp3 inflammasome activation [30], which in turn is mechanistically linked with sterile inflammation in DKD [31,32]. To scrutinize whether aPC regulates inflammasome activation in glucose-stressed tubular cells, we determined the mRNA expression of p65 NF-ĸB and inflammasome components (Nlrp3, caspase1, Asc and Il-1β) in BUMPT cells treated with high glucose for 24 h (Figure 4a). While the mRNA expression levels of p65 NF-ĸB remained unchanged, indicating posttranslational regulation of p65 NF-ĸB (e.g., via phosphorylation), the mRNA expression of Nlrp3, caspase1, Asc and Il-1β was markedly induced. Pretreatment of BUMPT cells with aPC (20 nM, 1 h) prevented the induction of inflammasome-related genes. Upon immunohistochemical analyses, the levels of phospho-p65 NF-ĸB, Nlrp3, Asc, and cleaved caspase-1 were all increased in the tubular compartment, corroborating the induction of the Nlrp3 inflammasome in the tubular compartment of diabetic mice (Figure 4b–f). Of note, upon immunohistochemical analyses, phospho-p65 NF-ĸB was increased, corroborating a posttranslational regulation of NF-ĸB, and was in particular observed in the nucleus (Appendix A, 2b-high-resolution images showing colocalization of p65 NF-ĸB signal with DAPI by ICORR, indicating nuclear localization). Inflammasome activation was not observed in diabetic APC^high^ mice, extending the findings made in BUMPT cells in vitro to the in vivo situation. Sterile inflammation is expected to trigger immune cell accumulation in the kidney. Indeed, the frequency of macrophages (F4/80+, immunohistochemistry) was increased in the kidneys of diabetic wild-type mice (Figure 4g,h). In agreement with inflammasome suppression in diabetic APC^high^ mice, no increase in renal macrophages (F4/80+) was observed in diabetic APC^high^ mice. Collectively, these data demonstrate that aPC maintains mitochondrial function and prevents tubular ROS formation and inflammasome activation in DKD (Figure 5).

## 4. Discussion

The pathogenesis of DKD is closely linked with sterile inflammation and mitochondrial dysfunction. We previously demonstrated that the anticoagulant and cytoprotective coagulation protease aPC protects against experimental DKD through receptor-dependent mechanisms, independently of its anticoagulant function [6,10,14,15,20,24]. In previous work, we focused mostly on the glomerular compartment, and we were able to demonstrate that aPC reduces mitochondrial ROS generation in glomerular cells. Here, we demonstrate that aPC likewise protects against glucose-induced increased ROS generation in tubular cells. Reduced tubular ROS generation in diabetic mice expressing a mutant protein C variant resulting in elevated plasma levels of aPC (APC^high^ mice) [6] was associated with reduced inflammasome activation and the reduced accumulation of macrophages in the tubular compartment. Taken together, these results establish that the cytoprotective coagulation protease aPC decreases glucose levels and induces excess ROS generation and sterile inflammation in the tubular compartment.

The cytoprotective effect of aPC is well documented, and it was approved as therapeutic in patients with sepsis between 2001 and 2009 [33,34]. While the precise cytoprotective mechanism remains unknown, aPC is known to target several signalling pathways and cellular functions. Among the latter, emerging evidence supports a role of aPC in the regulation of fundamental cellular processes such as the function of the endoplasmic reticulum [14] or of mitochondria [10]. Mitochondrial dysfunction is closely linked to excess generation of reactive oxygen species. In the context of DKD, we previously demonstrated that aPC reduces ROS generation in glucose-stressed podocytes, highly specialized epithelial cells of the renal glomeruli [10]. In podocytes, aPC suppresses the redox regulator p66^Shc^, thus reducing mitochondrial ROS generation. Likewise, aPC reduces glucose-induced ROS generation in atherosclerotic plaque-associated macrophages via p66^Shc^ [35]. Reduced ROS generation by aPC in monocyte-derived cells is in agreement with an earlier in vitro study [36,37], but the previous study used nonphysiological high concentrations of aPC and failed to provide a mechanism. Whether aPC reduces mitochondrial ROS via p66^Shc^ in tubular cells remains to be shown. Furthermore, whether aPC directly regulates p66^Shc^ or the inhibition of p66^Shc^ simply compensates for excess ROS generation in the absence of aPC remains to be shown.

In the current study, aPC regulates a number of genes related to the mitochondrial electron-transport chain. This observation suggests that the effect of aPC on mitochondrial ROS generation is not restricted to the regulation of p66^Shc^. Congruently, levels of PGC1a, a master regulator of mitochondrial biogenesis, were maintained in diabetic mice with increased levels of aPC, suggesting that aPC may maintain mitochondrial function through an undefined higher-level regulator mechanism. A more profound effect of aPC on mitochondrial function is supported by observations made in mice with low levels of aPC. Mice carrying a point mutation in thrombomodulin (TMPro), the cofactor required for thrombin-dependent protein C activation, display altered mitochondrial morphology (increased mitochondrial branching), impaired mitochondrial metabolism, altered cardiolipin composition and accumulation of the D17 mitochondrial loop deletion in cells of the central nervous system [38]. Intriguingly, in TMPro mice, the expression of components of the mitochondrial electron transport chain in the brain is altered, mirroring observations made in the current study [38]. Taken together, these data support that the primarily endothelial-dependent thrombomodulin protein C system maintains mitochondrial function and ROS generation through a currently poorly defined pathway. Importantly, this endothelial-dependent pathway maintaining mitochondrial function appears to be disrupted in the setting of diabetes mellitus and diabetes-associated endothelial dysfunction. Defining the underlying mechanism may uncover new approaches to targeting and maintaining mitochondrial function in diabetic patients.

The current results additionally suggest a new pathway that promotes inflammasome activation in the setting of diabetes mellitus. Loss of thrombomodulin-mediated protein C activation in the setting of myocardial or renal ischemia reperfusion injury has been previously linked with NLRP3 inflammasome activation [32], but whether impairment of the endothelial thrombomodulin protein C system likewise promotes inflammasome activation during a chronic disease remains unknown. Activation of the NLRP3 inflammasome in the setting of diabetes mellitus and specifically in DKD has likewise been shown before [30,31,39,40] The current data support the concept that endothelial dysfunction with impaired aPC generation promotes NLRP3 inflammasome activation and sterile inflammation in DKD. These data raise the intriguing possibility that targeting aPC signalling, possibly using small molecules that mimic biased aPC signalling via protease-activated receptor 1 such as parmodulin [32,41], may reduce inflammasome activation, as observed in the setting of ischemia–reperfusion injury. Importantly, as NLRP3 inhibition by aPC is independent of its anticoagulant effect [32], such therapy seems in principle feasible in the long term without increasing the risk of haemorrhage.

The current study, like any study, has limitations. First, we used only a mouse model, and whether the mechanisms proposed here are the same in humans remains to be shown. Furthermore, we used only a mouse model for type 1 diabetes. However, as aPC exerts nephroprotective effects in both type 1 and type 2 diabetes models [6,10,14,15] and reduces inflammasome activation in other diseases [32,42], we are confident that the current findings can be extended to type 2 diabetes.

In summary, we provide new evidence for the regulation of mitochondrial function by the cytoprotective coagulation protease aPC. Whether aPC likewise modulates other functions of mitochondria, such as mitochondrial metabolism and metabolites, remains to be shown.

## Figures and Tables

**Figure 1 nutrients-14-03138-f001:**
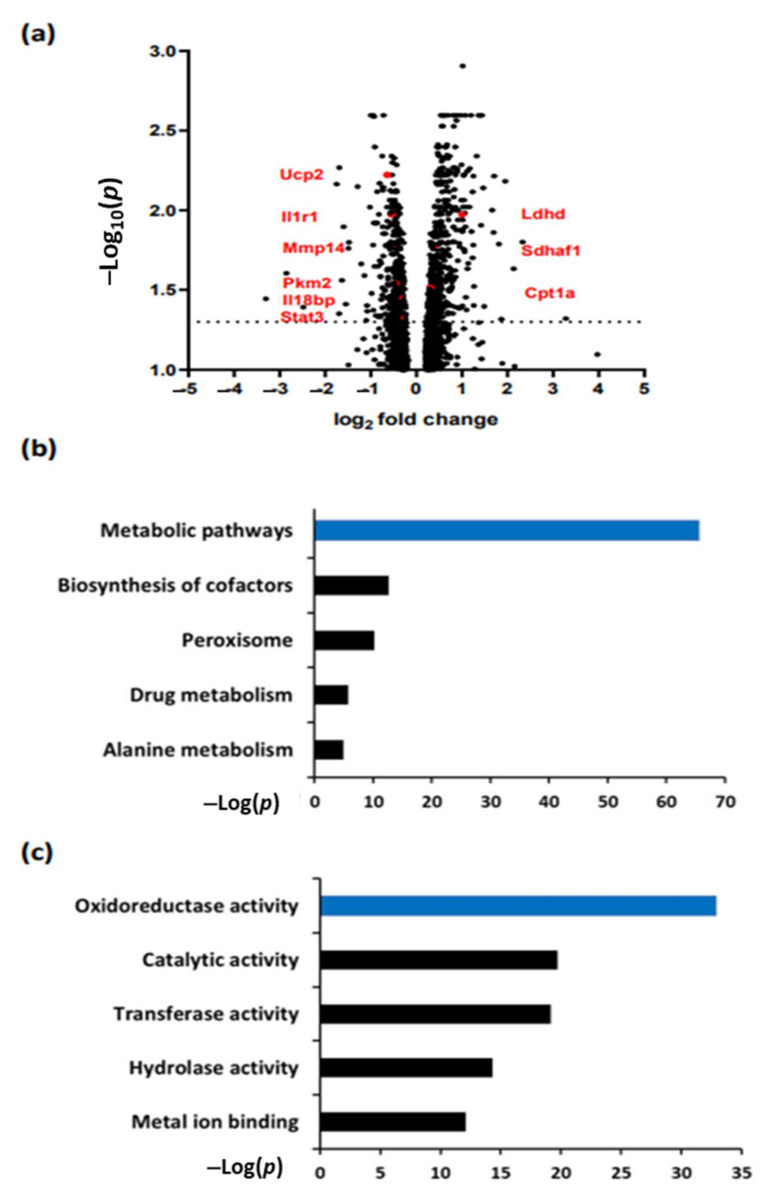
aPC regulates metabolic pathways and oxidoreductase activity in DKD. (**a**) Volcano plot showing differentially expressed genes (DEGs) in APC^high^-DM mice compared with WT-DM mice. Example genes related to metabolism and inflammatory pathways are shown in red (bold). (**b**) KEGG pathway analysis of renal (induced) DEGs enriched by DAVID in APC^high^-DM mice compared to WT-DM mice. The bar plot shows the enrichment scores (-log [*p* value]) of the significantly enriched KEGG pathways. (**c**) KEGG pathway analysis of metabolism-related DEGs enriched by DAVID in APC^high^-DM mice compared to WT-DM mice. The bar plot shows the enrichment scores (−log [*p* value]) of the significantly enriched KEGG pathways.

**Figure 2 nutrients-14-03138-f002:**
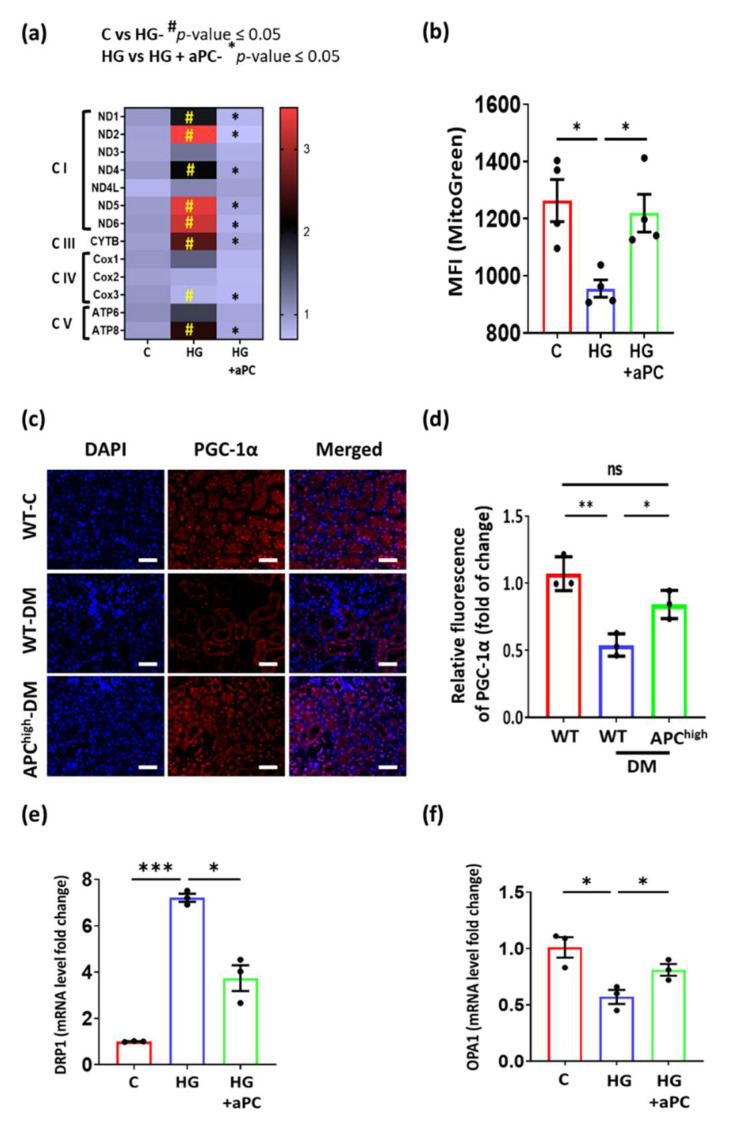
aPC protects against mitochondrial dysfunction in DKD. (**a**) Heatmap summarizing the mRNA expression (quantitative RT–PCR) of genes related to mitochondrial ETC complexes. Five samples per group were analysed, and the colour key represents the fold change. (**b**) Bar graph with dot-plot summarizing staining intensity (mean fluorescent intensity, MFI) of the mitochondrial mass marker MitoTracker Green in tubular cells without (C) or with (HG) exposure to high glucose (25 mM, 24 h). A subgroup of cells was pretreated with aPC (20 nM, 1 h: HG + aPC). (**c**,**d**) Representative immunohistochemical images (**c**) and bar graph with dot plot summarizing results (**d**) of PGC-1α staining (red, DAPI nuclear counterstain, blue) in mouse kidneys from nondiabetic wild-type (WT-C), diabetic wild-type (WT-DM), or diabetic APC^high^ mice (APC^high^-DM), scale bar: 20 µm. (**e**,**f**) Bar graph with dot-plot summarizing Drp1 (**e**) and Opa1 (**f**) mRNA levels in control (normal glucose, 5 mM) and high glucose (25 mM)-treated BUMPT cells without (HG) or with aPC pretreatment (20 nM, 1 h, HG + aPC); * *p* < 0.05; ** *p* < 0.01; *** *p* < 0.001; ns = non-significant (**a**,**b**,**d**–**f**). Statistical analyses were performed using one-way ANOVA with Sidak’s multiple comparison test. aPC = activated protein C; C = control; DM = diabetes mellitus; DRP1 = dynamin-related protein 1; HG = high glucose; MFI = mean fluorescent intensity; OPA1 = Optic atrophy 1; WT = wild-type.

**Figure 3 nutrients-14-03138-f003:**
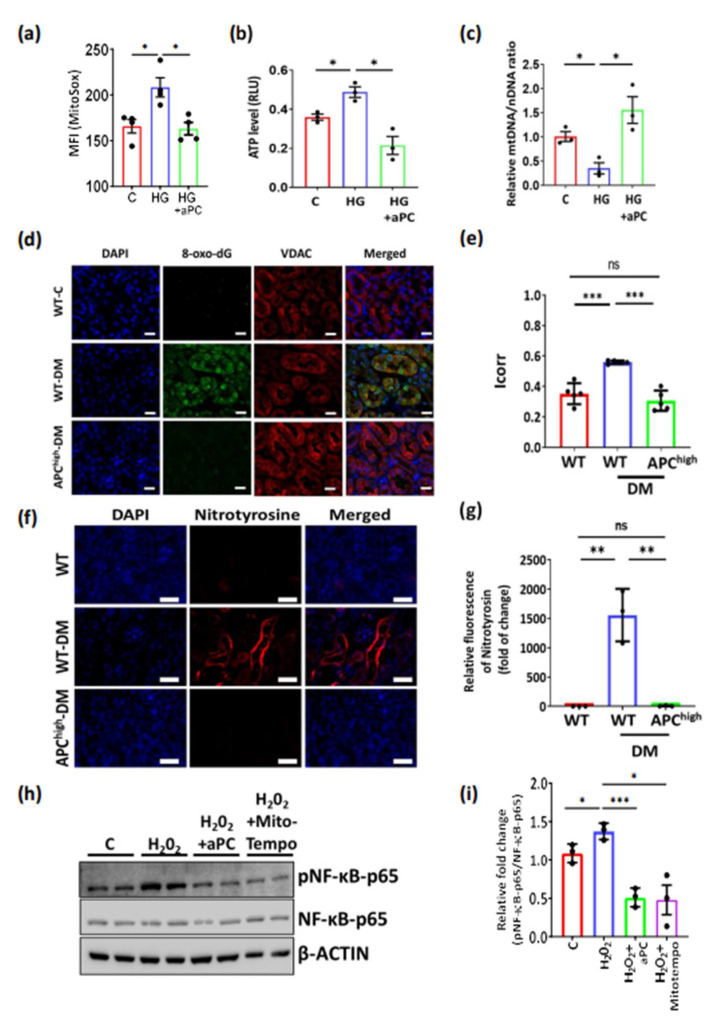
aPC reduces tubular ROS in diabetic nephropathy. (**a**) Bar graph with dot-plot summarizing staining intensity (mean fluorescent intensity, MFI) of the mitochondrial ROS marker MitroSox in tubular cells without (C) or with (HG) exposure to high glucose (25 mM, 24 h). A subgroup of cells was pretreated with aPC (20 nM, 1 h: HG + aPC). (**b**) ATP levels were measured using a luciferin-luciferase assay in control (normal glucose, 5 mM) and high glucose (25 mM)-treated BUMPT cells without (HG) or with aPC pretreatment (20 nM, 1 h, HG + aPC); RLU (relative light unit) after normalization for cell number of each sample. (**c**) Relative quantification was performed using qRT-PCR by amplification of ND2 and normalized against the hexokinase gene (HK). (**d**,**e**) Representative images of immunohistochemical staining of 8-oxo-dG and VDAC and dot plots showing Icorr scores for 8-oxo-dG and VDAC colocalization in mouse kidneys obtained from nondiabetic wild-type (WT-C), diabetic wild-type (WT-DM), or diabetic APC^high^ mice (APC^high^-DM); scale bar: 20 µm. (**f**,**g**) Representative images of the immunohistochemical staining of nitrotyrosine (red; DAPI nuclear counterstain, blue) in the tubular compartment of mouse kidney; groups as in c and (**g**) bar graphs with dot summarizing results. (**h**,**i**) Representative immunoblots and (**i**) bar graphs with dot plots summarizing the data of phospho-p65-NF-ĸB in BUMPT cells stimulated by H_2_O_2_ (H_2_O_2_, 100 µM/24 h) or with pretreatment with aPC (20 nM, 1 h) or Mito-Tempo (10 µM, 1 h); nonphosphorylated p65 NF-ĸB as a control, β-actin: loading control. * *p* < 0.05; ** *p* < 0.01; *** *p* < 0.001 (**a**–**c**,**e**,**g**,**i**). Statistical analyses were performed using one-way ANOVA with Sidak’s multiple comparison test. aPC = activated protein C; ATP = adenosine 5′-triphosphate; C = control; DAPI = 4′,6-diamidino-2-phenylindole; DM = diabetes mellitus; HG = high glucose; MFI= mean fluorescent intensity; VDAC = voltage-dependent anion channel; WT = wild-type; 8-oxo-dG = 8-Oxo-2’-deoxyguanosine.

**Figure 4 nutrients-14-03138-f004:**
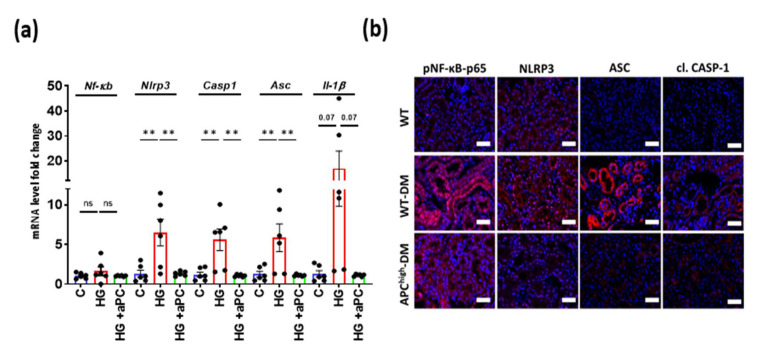
aPC reduces inflammasome activation in hyperglycaemia. (**a**) Bar graph with dot plot summarizing mRNA levels of p65 NF-ĸB, Nlrp3, Asc, and Casp1 (q-RT–PCR) in control (normal glucose, 5 mM) and high glucose (25 mM)-treated BUMPT cells without (HG) or with aPC pretreatment (20 nM, 1 h, HG + aPC). (**b**–**f**) Representative immunofluorescence images (**b**) and bar graphs with dots summarizing the results (**c**–**f**) of kidney sections stained for pNf-ĸb-p65, Nlrp3, cleaved caspase-1 or Asc (all red, DAPI nuclear counterstain, blue; groups as described in a. (**g**,**h**) Representative immunofluorescence images (**g**) and bar graphs summarizing the results (**h**) of kidney sections stained for macrophages (F4/80, red; DAPI nuclear counterstain, blue); arrow indicates positive staining. * *p* < 0.05; ** *p* < 0.01; *** *p* < 0. 001 (**a**,**c**–**f**,**h**). Statistical analyses were performed using one-way ANOVA with Sidak’s multiple comparison test. aPC = activated protein; ASC = adaptor molecule apoptosis-associated speck-like protein containing a CARD; C = control; Casp1 = Caspase- 1; DAPI = 4′,6-diamidino-2-phenylindole; DM = diabetes mellitus; HG = high glucose; MFI= mean fluorescent intensity; NF-κB = Nuclear factor kappa-light-chain-enhancer of activated B cells; NLRP3 = NLR family pyrin domain containing 3; VDAC = voltage-dependent anion channel; WT = wild-type.

**Figure 5 nutrients-14-03138-f005:**
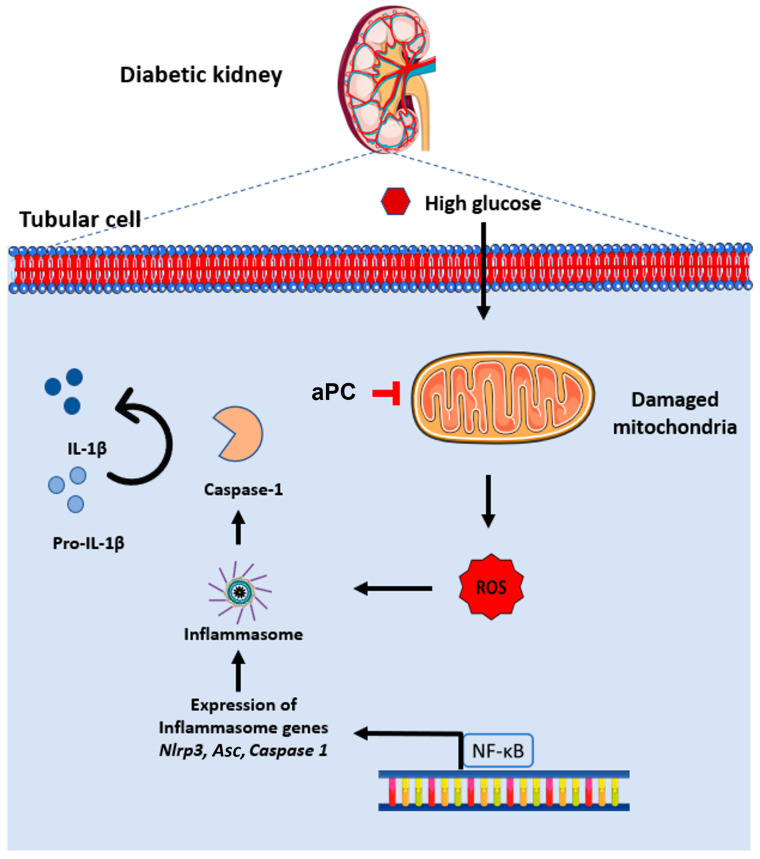
Visual abstract: aPC protects the kidney from high glucose-induced ROS-dependent inflammasome activation. aPC = activated protein; ASC = adaptor molecule apoptosis-associated speck-like protein containing a CARD; Casp1 = Caspase- 1; IL-1β = Interleukin-1 β; NF-κB = Nuclear factor kappa-light-chain-enhancer of activated B cells; NLRP3 = NLR family pyrin domain containing 3; ROS = reactive oxygen species.

## Data Availability

Excluded.

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
