# Peer review of "Activated Protein C Ameliorates Tubular Mitochondrial Reactive Oxygen Species and Inflammation in Diabetic Kidney Disease"

_nutrients, 2022, doi:10.3390/nu14153138_

Round 1

Reviewer 1 Report

In the manuscript entitled, Activated protein C ameliorates tubular mitochondrial 2 reactive oxygen species and inflammation in diabetic 3 kidney disease, the authors provided new insights in Diabetic kidney disease (DKD). Briefly, the authors described how the endothelial-dependent cytoprotective coagulation protease activated protein C (aPC) ameliorates the glucose-induced mitochondrial ROS production in tubular cells and dampens the associated renal sterile inflammation.

Figure 1: Supplementary Figures and Figure 1 is missing. A better description of the statistical parameters used for the RNA-seq analysis.

Figure 2: To better describe the mitochondrial health in case or not of external damage, the mitochondrial respiration assay should be performed in vitro using all the experimental conditions used in the manuscript. In addition, mitochondrial DNA (mtDNA) content and ATP levels should be determined.

A better quality of the images and a better description of how MFI and PGC1alpha intensity were performed are required.

Can the author hypothesize whether the proximal, distal tubule or both are more affected or involved in the aPC effect?

Figure 3: from the Figure 3, high 8-oxo-dg staining is present in the nuclei of APChigh mice. Did the authors check any damage in the nucleus of tubular cells?

Figure 4: an increase of ROS can induce apotosis or mitophagy. Did author check any aspects of these two other mechanisms of cell death?

Reviewer 2 Report

Activated protein C (aPC) has been reported to have a protective effect in diabetic kidney disease (DKD). The current manuscript (1825456) by Rana et al. further confirmed the beneficial effects of aPC in tubular cells. Using the APChigh mice, it is shown that aPC regulates the genes involved in mitochondrial biogenesis and energy metabolism. Furthermore, it decreases mitochondrial ROS levels, and reduces inflammation. For most part, the study was well designed, and the data was convincing. My concern is that the current study is an incremental step from their prior research. It fails to significantly expand the current knowledge on aPC. It is recommended that the manuscript be published elsewhere. 

Round 2

Reviewer 1 Report

The Authors have addressed my concerns and have made corrections in the text. The manuscript can be accepted for publication.

Reviewer 2 Report

The authors have properly addressed my concerns. I recommend the manuscript be accepted in its current form.